# Apolipoproteins—New Biomarkers of Overweight and Obesity among Childhood Acute Lymphoblastic Leukemia Survivors?

**DOI:** 10.3390/ijms231810634

**Published:** 2022-09-13

**Authors:** Klaudia Sztolsztener, Hubert Żywno, Katarzyna Hodun, Katarzyna Konończuk, Katarzyna Muszyńska-Rosłan, Eryk Latoch

**Affiliations:** 1Department of Physiology, Medical University of Bialystok, 15-222 Białystok, Poland; 2Department of Pediatric Oncology and Hematology, Medical University of Bialystok, 15-274 Białystok, Poland

**Keywords:** apolipoproteins, survivors, overweight, childhood acute lymphoblastic leukemia, pediatric cancer

## Abstract

Patients suffering from childhood acute lymphoblastic leukemia (ALL) are at risk of late adverse treatment-related effects. The examination of targeted biomarkers could be used to improve the diagnosis and prediction of life-threatening ALL sequelae. The purpose of this cross-sectional study was to search for treatment-related alterations in apolipoprotein (Apo) levels as potential markers of the occurrence of obesity in subjects treated for ALL, and to assess the relationships between weight, gender, anticancer treatment, and Apo concentrations. Fifty-eight ALL survivors were included in the study. The mean time of follow-up after treatment cessation was 5.41 ± 4.29 years. Serum levels of apolipoproteins were measured using a multiplex assay kit. Among ALL survivors, we observed a significant correlation of Apo-C1, Apo-C3, Apo-H, and Apo-J levels, depending on body mass index (BMI). Marked differences were observed in the area under the curve of Apo-A1, Apo-A2, Apo-C1, Apo-D. In our study, patients with a history of childhood ALL developed alterations in their Apo profile. Furthermore, this is the first study revealing that some apolipoproteins may act as valuable biomarkers useful in the prognosis of metabolic imbalance. We believe that this paper, at least partially, will highlight the importance of long-term prognosis of metabolic complications associated with the anticancer chemotherapy used to treat hematological malignancies in children.

## 1. Introduction

Today, acute lymphoblastic leukemia (ALL) is the most common childhood cancer, accounting for 75% of leukemias and 25% of all cancers [1,2]. Despite the significant improvement in the survival of patients suffering from childhood ALL over the past few decades, surviving patients are at risk of a spectrum of late adverse treatment-related effects [3]. The data show that about 50% of childhood cancer survivors (CCS) experience at least one endocrine disruption, which is more frequent than in their healthy peers [4,5]. Numerous studies have shown that ALL survivors often struggle with premature onset of many disorders, such as obesity, dyslipidemia, hypertension (HT), and metabolic syndrome (MetS) [6,7]. According to an available meta-analysis based on 47 studies, it should be noted that obesity was present in 46% of the recovered patients during the 10 years of follow-up [8]. In addition, one of the crucial criteria of metabolic syndrome—i.e., the abnormal distribution of body fat (visceral obesity) observed in ALL survivors—leads to various dysfunctions, especially in lipid metabolism [9].

The childhood survivors of ALL constitute a group of patients with a particularly high risk of the development of late effects due to anticancer treatment. Anticancer therapy is radical and intense, including cytostatic agents, high doses of glucocorticoids and, in selected cases, total body irradiation (TBI) and/or cranial radiotherapy (CRT) [8,10]. It is known that the abovementioned types of treatment may affect multiple metabolic pathways, in some cases resulting in lipid disturbances [11]. Irradiation of the hypothalamus can disrupt, among other things, the appetite regulation center, leading to hyperphagia and progressive obesity [12,13]. Moreover, glucocorticoids and chemotherapeutics affect both carbohydrate and lipid metabolism, contributing to many side effects, including obesity [6,14].

It has been shown that the diagnosis of late sequelae of ALL is often delayed, which has a negative impact on the overall health status and further prognosis of the survivors [15,16]. Thus, the examination of targeted biomarkers can be used to improve the diagnosis and prediction of the life-threatening sequelae of ALL. It seems that apolipoproteins (Apo) may be new potential indicators for the improvement of diagnosis and prediction of follow-up treatment complications in ALL survivors [11]. Apolipoproteins play a crucial role in lipid metabolism and transport, whereby their levels could have a relevant contribution to the development of obesity and other metabolic disorders. These proteins are a structural component of lipoproteins that exert effects on enzyme activity engaged in lipid metabolism, regulating the lipid balance. Apolipoproteins differ in their biological properties and clinical functions [17,18]. Apolipoprotein A1 (Apo-A1) is involved in the reverse transport of cholesterol from the peripheral tissues to the liver [17,19]. In contrast, apolipoprotein A2 (Apo-A2) exhibits an atherogenic effect. Interestingly, Apo-A2 is considered to be a strong predictor of metabolic syndrome and a risk factor in the occurrence of cardiovascular diseases [20]. In addition, apolipoproteins C1, C2, and C3 (Apo-C1, Apo-C2, and Apo-C3) regulate endothelial cell lipoprotein lipase (LPL) activity, and directly influence the levels of triacylglycerol (TAG) released into the circulation [21]. Some evidence indicates that the dysfunction of LPL activity leads to hypertriglyceridemia, contributing to the development of obesity [22,23].

Abnormalities in Apo levels are associated with the development of metabolic syndrome components [24]. The mechanisms underlying the development of a spectrum of disorders in lipid metabolism in survivors of acute lymphoblastic leukemia are still unclear. Thus, the purpose of this cross-sectional study was to examine changes in the serum content of a broad panel of nine apolipoproteins in ALL survivors. It is essential to search for treatment-related alterations in adipose tissue metabolism and the potential markers of the occurrence of obesity in patients treated for ALL in the past, and to assess the relationships between weight, gender, anticancer treatment, and Apo concentrations.

## 2. Results

### 2.1. Characteristics of the Study Group

Table 1 presents the characteristics of the acute lymphoblastic leukemia (ALL) survivors. In the whole study group, the mean body mass index (BMI) was 21.95 ± 5.29 kg/m^2^. All of the patients were classified into one of three groups in terms of weight status: normal weight (53.45%), overweight (29.31%), and obese (17.24%).

### 2.2. The Comparison of Apolipoprotein Concentrations between ALL Survivors (With Normal and Abnormal Weight) and Control Patients

The data analysis of Apo concentrations within ALL study participants and the control group is presented in Figure 1. We observed that apolipoprotein A1 (Apo-A1) and apolipoprotein A2 (Apo-A2) levels were significantly higher among ALL survivors compared to the control group (259.55 ± 125.56 vs. 135.25 ± 67.16 mg/dL and 24.75 ± 6.39 vs. 18.00 ± 8.77 mg/dL, respectively; *p* < 0.05; Figure 1A,B). The opposite effect was observed in apolipoprotein D (Apo-D) concentration, which was lower in childhood cancer survivors than in the control group (3.66 ± 0.57 vs. 17.88 ± 11.71 mg/dL; *p* < 0.05; Figure 1F). Additionally, C-reactive protein (CRP) concentration were considerably increased in childhood cancer survivors compared to the control group (0.56 ± 0.23 vs. 0.38 ± 0.13 mg/dL; *p* < 0.05; Figure 1J).

As shown in Table 2, overweight and obese ALL survivors had lower concentration of apolipoprotein C1 (Apo-C1) and higher concentration of CRP compared to the ALL patients with normal weight (18.37 ± 5.59 vs. 22.35 ± 6.97 mg/dL and 1.17 ± 1.10 vs. 0.58 ± 0.38 mg/dL, respectively; *p* < 0.05; Table 2). The analysis by BMI revealed that ALL survivors with normal weight had higher concentrations of Apo-A1, Apo-A2, and apolipoprotein J (Apo-J), and lower concentration of apolipoprotein D (Apo-D), compared to the control patients with normal body weight (Apo-A1: 263.93 ± 139.68 vs. 135.25 ± 67.16 mg/dL; Apo-A2: 25.34 ± 6.07 vs. 18.00 ± 8.78 mg/dL; Apo-J: 11.45 ± 3.70 vs. 9.59 ± 2.88 mg/dL; Apo-D: 3.63 ± 0.72 vs. 28.74 ± 28.90 mg/dL; *p* < 0.05; Table 2). We also observed that in overweight and obese ALL patients the concentrations of Apo-A1, Apo-A2, and CRP were increased compared to the control patients with normal weight (Apo-A1: 254.32 ± 108.86 vs. 135.25 ± 67.16 mg/dL; Apo-A2: 24.04 ± 6.80 vs. 18.00 ± 8.78 mg/dL; CRP: 1.17 ± 1.10 vs. 0.65 ± 0.81 mg/dL; *p* < 0.05; Table 2). Moreover, Apo-C1 and Apo-D levels were decreased in the overweight and obese subgroup from the ALL group in comparison with the control patients with normal weight (18.37 ± 5.59 vs. 24.31 ± 9.04 mg/dL and 3.60 ± 0.61 vs. 28.74 ± 28.90 mg/dL, respectively; *p* < 0.05; Table 2).

### 2.3. The Concentration of Apolipoproteins in ALL Survivors, Taking into Account Gender, Age at Diagnosis, and Radiotherapy

The 58 qualified patients consisted of 27 females and 31 males, and showed greater level of Apo-A1 in females than in males, but this difference did not achieve statistical significance (*p* = 0.111; Appendix A).

The comparison of Apo concentrations according to the age at diagnosis of ALL is presented in Table 3, but we did not observe any significant differences.

The comparison of selected parameters according to the history of radiotherapy is presented in Appendix A. Among all of the parameters, only the levels of triacylglycerol (TAG), apolipoprotein C3 (Apo-C3), and Apo-J were slightly lower in patients without radiotherapy, but these differences were not statistically significant (*p* = 0.059, *p* = 0.127, and *p* = 0.114, respectively; Appendix A).

### 2.4. The Distribution of Apolipoprotein Concentrations in ALL Survivors by Weight Status

The Spearman’s rank correlation estimates for the ALL group by weight status are presented in Table 4. We found a moderate negative correlation between Apo-C1 concentration and normal weight in ALL survivors (Apo-C1: r = −0.59; *p* < 0.05). Additionally, we also observed that overweight and obesity among ALL survivors were positively correlated with the levels of Apo-C3, apolipoprotein H (Apo-H), Apo-J, and CRP (Apo-C3: r = 0.60; Apo-H: r = 0.49; Apo-J: r = 0.41; CRP: r = 0.47; *p* < 0.05; ).

### 2.5. The Analysis of Apolipoprotein Concentrations for the Prediction of the Occurrence of Overweight and Obesity in ALL Survivors

Based on the Apo concentrations, the receiver operating characteristic curves to predict overweight and obesity in childhood cancer survivors were calculated (Figure 2). Statistically significant differences (*p <* 0.05) in the area under the curve (AUC) of apolipoprotein profiles were observed in Apo-A1 (AUC 0.80), Apo-A2 (AUC 0.68), Apo-C1 (AUC 0.68), Apo-D (AUC 0.87), and CRP (AUC 0.90), as presented in Figure 2. The results indicate that Apo-D had a great AUC value, and may be a predictor of overweight and obesity in the studied participants. Moreover, there were no alterations in the other analyzed variables (apolipoprotein parameters) for predicting unhealthy weight status in ALL survivors.

### 2.6. The Correlation of Apolipoprotein Concentrations with Selected Basic Parameters in ALL Survivors

The multiple Spearman’s rank correlations were performed regarding the selected characteristic features of all of the patients with ALL (Appendix A). We observed that Apo-C3, Apo-D, Apo-H, Apo-J, and CRP were moderately-to-strongly correlated with BMI (Apo-C3: r = 0.60; Apo-D: r = 0.56; Apo-H: r = 0.49; Apo-J: r = 0.41; CRP: r = 0.47; *p* < 0.05; Appendix A). We also noted that Apo-C3 was related to SBP (Apo-C3: r = 0.46; *p* < 0.05; Appendix A) and Apo-C1 was related to DBP (Apo-C1: r = 0.46; *p* < 0.05; Appendix A) to moderate degrees in the study participants. Moreover, this analysis did not show any associations in the apolipoprotein panel between WHR and age at ALL diagnosis (Appendix A).

### 2.7. The Correlation of Apolipoprotein Concentrations with Selected Treatment-Related Factors in ALL Survivors

Table 5 presents the Spearman’s rank correlation of apolipoprotein concentrations in the ALL group depending on the selected type of anticancer chemotherapy, except for prednisone treatment (all patients received the same dose of this glucocorticoid). There was no significant correlation between Apo concentrations and the use of anticancer chemotherapy (Table 5).

## 3. Discussion

The treatment of ALL during childhood, involving anticancer chemotherapeutics, irradiation, and corticosteroid therapy (e.g., dexamethasone and prednisone), is highly associated with the development of MetS components—especially abnormal alterations in lipid profiles, such as hypertriglyceridemia and hypercholesterolemia [25,26]. Numerous studies have shown that pediatric ALL survivors display an altered lipidomic profile, involving increased levels of plasma TAG and low-density lipoprotein cholesterol (LDL-C), whereas high-density lipoprotein cholesterol (HDL-C) level is decreased. These changes are often linked with the presence of overweight and obesity [11,27]. The possible mechanisms underlying this metabolic imbalance are poorly understood. However, it may be correlated with the intensified hepatic synthesis of TAG and its impaired hydrolysis in adipose tissue due to inhibition of LPL activity [28]. To the best of our knowledge, this is the first study aiming to evaluate whether serum Apo distribution is deteriorated by anticancer chemotherapeutics in a weight-dependent manner, as well as showing an in-depth analysis of the correlation between Apo composition, body mass index (BMI), and anticancer treatment used among pediatric ALL survivors.

In our study, significantly increased serum concentrations of Apo-A1 and Apo-A2 were observed in the whole ALL group—which included participants with normal weight, overweight, and obesity—in comparison with healthy peers. The main roles of Apo-A1 are the removal of cholesterol from cells and enhancing the activity of lecithin–cholesterol acyltransferase (LCAT), which results in efficient reverse plasma cholesterol transport. In turn, Apo-A2 regulates LPL activity through HDL-C proteasome modulation [29,30]. Thus, our results suggest that ALL survivors may be at risk of diminished lipid turnover induced by the anticancer treatment. Experiments on Apo-A2 transgenic rabbits and clinical studies involving patients with coronary artery disease, atherosclerosis, or hypercholesterolemia have shown that increased serum concentration of Apo-A2 may constitute an important contributor to the development of MetS [31,32,33]. The study conducted by Baroni et al. noted that the serum concentration of Apo-A1 was decreased in ALL patients. The same investigation showed that Apo-A1 content was markedly elevated after the induction phase of anticancer treatment. These findings are consistent with our results, indicating a direct correlation between the serum level of Apo-A1 and ALL treatment [34]. In contrast, Parsons et al. reported a decrease in the level of Apo-A1 after therapy in both short- and long-term survivors [35]. Moreover, a study conducted by Morel et al. revealed that pediatric ALL survivors had decreased Apo-A1 level in parallel with elevated concentrations of Apo-A2, Apo-B100, and Apo-C3 [11]. The causes of these discrepancies may lie in the different follow-up periods and various features of the examined groups of patients, which involved only individuals with normal BMI. Furthermore, in the present study, overweight or obese patients demonstrated significantly decreased level of Apo-C1 compared to the group of normal-weight ALL survivors. Importantly, Apo-C1 is a potent regulator of LPL, hepatic lipase (HL), and LCAT activities, thereby playing an important role in efficient lipid transport and metabolism [36]. Interestingly, Gautier et al. showed that unaltered expression of Apo-C1 in transgenic rabbits attenuated the progression of atherosclerosis by reducing lipid oxidation and cholesteryl ester transfer protein (CETP) activity [37]. The results of our study also showed that the concentration of Apo-D was decreased in the whole ALL group in comparison with the control subjects. Decreased level of Apo-D may be correlated with the reduced effective very-low-density lipoprotein–triacylglycerol (VLDL–TAG) clearance via interaction with low-density lipoprotein receptor-related protein (LRP), leading to impaired lipid metabolism [38]. Moreover, our study revealed that children with a history of ALL affliction presented notably increased CRP concentration in comparison with the control group. Elevated level of CRP among ALL survivors was associated with an increased risk of cardiometabolic complications, including obesity, dyslipidemia, and insulin resistance [39]. Furthermore, CRP itself may also significantly affect the Apo composition, thereby leading to disturbances in Apo profiles. This may constitute another mechanism of metabolic imbalance that is widely present among childhood ALL survivors. It will be necessary to clarify the link between CRP and Apo in future studies. In general, the presence of the abovementioned discrepancies indicates that changes in Apo profiles induced by anticancer treatment in ALL survivors might also result from interindividual changes in lipid metabolism.

An important question is whether CRT—another commonly used treatment mode—might cause lipid abnormalities among ALL survivors. CRT is widely known for the induction of long-term side effects in ALL survivors, including impairment within the hypothalamic–pituitary (HP) axis and cardiovascular complications [40], although we did not observe any significant effects of this procedure on Apo profiles in our study. Among the examined subjects, only a trend towards increases in the concentrations of TAG, Apo-C3, and Apo-J was observed. Our results are consistent with those of the study conducted by Mohaptera et al., who showed that CRT was not a risk factor for the development of MetS among ALL subjects [6]. Another explanation for this may be the use of much lower radiotherapy (RT) doses in patients treated for ALL compared to brain tumors, as doses for ALL rarely exceed 12 Gy. However, it should be noted that cranial radiation with higher doses leads to the occurrence of at least one HP dysfunction in 51% of cases, and may be associated with weight gain, as a factor of MetS in ALL survivors [41,42,43]. Therefore, these contradictory results might stem from a very small group of individuals who underwent cranial radiation with an insufficient follow-up period.

It is important to note that the alteration in lipid profiles—especially in the composition of lipoproteins—can be a prognostic factor in patients with active leukemia, and may also be a reliable marker of remission in the long-term effects on childhood ALL survivors [34,44]. In our study, receiver operating characteristic (ROC) analysis revealed that serum concentrations of Apo-A1, Apo-A2, Apo-C1, and Apo-D might act as specific and sensitive biomarkers for the prediction of the development of overweight and obesity in subjects after the cessation of anticancer therapy. Furthermore, Spearman’s correlation analysis revealed that there is a link between BMI and the levels of Apo-C3, Apo-D, Apo-H, and Apo-J, as well as CRP, among ALL survivors. Increased levels of Apo-C3 and Apo-H induce inhibition of triacylglycerol-rich lipoprotein (TRL) lipolysis and impair the hepatic TRL clearance, which may result in hypertriglyceridemia and the formation of atherosclerotic plaques, thereby significantly increasing the risk of cardiovascular disorders [45,46]. Moreover, Won et al. showed that increased level of Apo-J among adults with high BMI was correlated with systemic inflammation, as well as the development of MetS [47]. In our study, we did not find any direct correlations between altered serum Apo concentrations and the anticancer drugs used. However, it has been previously reported that ALL treatment—especially during the induction phase—may substantially affect the Apo profile among pediatric patients, while emphasizing the poorly understood mechanism of these findings [34,35]. Thus, the potential influence of the anticancer treatment on the concentration of various apolipoproteins should be further investigated. It should be noted that there are no studies showing the potential role of apolipoproteins in the occurrence of MetS components in the long-term prognosis of ALL survivors. Therefore, this research area is worthy of attention, and should be continued in future research.

There are several limitations to this study. First, it was a single-center analysis with a relatively small number of participants. It should be noted that apolipoproteins are molecules that guide the formation of lipoproteins, as well as acting as activators and inhibitors of enzymes involved in the lipoproteins’ metabolism [48]. Alterations in Apo profiles, as the important structural component of lipoproteins (we did not examine HDL-C, LDL-C, or VLDL-TAG levels—a significant limitation of this study), may be associated with disturbances within the lipoprotein profile, which might be related to the occurrence of many metabolic disorders, and could be established an important factor in cardiovascular complications. Moreover, this study could be supplemented with a comparison of Apo levels between ALL survivors with abnormal weight and overweight and obese children from a control group, which was not included in this cross-sectional study. The strengths of our study include pooled testing of nine apolipoproteins, the homogenous group of ALL patients, the relatively long follow-up period, and the absence of ethnic diversity.

## 4. Materials and Methods

### 4.1. Study Population and Anthropometric Measurements of Body Composition

In total, 58 (27 female and 31 male) White childhood cancer survivors (CCS) in complete continuous remission were involved in the study. All of the participants were diagnosed with ALL, and were treated at the Department of Pediatric Oncology and Hematology of the Medical University of Bialystok in Poland. The mean age at diagnosis was 5.01 ± 3.46 years. Additionally, the mean time of follow-up after cessation of treatment was 5.41 ± 4.29 years. The basic characteristics of the study group are shown in Table 1. The treatment was conducted according to the protocols of the International Berlin–Frankfurt–Münster Group (I-BFM), approved by the Polish Pediatric Leukemia/Lymphoma Study Group. According to the international treatment protocols, all of the patients were treated with cytostatic and glucocorticoid drugs. The means of each anticancer agent dose received by the patients with ALL are presented in detail in Table 1. Radiation therapy (RT) was used for the treatment of 9 patients, 8 of whom received irradiation of the central nervous system, while 2 individuals received TBI. In addition, in 6 patients diagnosed with ALL, allogeneic hematopoietic stem-cell transplantation (allo-HSCT) was performed. All of the parents signed participation forms before being enrolled in the study. The control group consisted of 22 healthy peers (female and male), who were the offspring of the clinic staff. In addition, the volunteers had normal body weight and proper BMI, and they did not receive any medication at the time of the study. All of the data concerning the performed examinations, treatment, body weight, sex, age, and type of diagnosis were recorded in the medical database. The study was performed in accordance with the principles of the Declaration of Helsinki, and was approved by the Ethical Committee of the Medical University of Bialystok (No. R-I-002/463/2016).

During the follow-up visit, all of the patients underwent anthropometric measurements and clinical examinations. Their height and weight were measured using a Martin anthropometer and a digital scale (Seca, Hamburg, Germany), respectively. In order to calculate BMI, the weight expressed in kilograms was divided by height expressed in squared meters (kg/m^2^). In accordance with the OLA/OLAF centile charts of BMI for Polish children, individuals were assigned to particular weight groups, i.e., normal weight, overweight, or obese. The BMI indicative of overweight was considered when the value exceeded +1 standard deviation (SD), while obesity was considered when the value exceeded +2 SD [49,50]. The waist–hip ratio (WHR) was calculated by waist circumference divided by hip circumference. Abdominal obesity in girls was defined when the value of WHR was greater than 0.8, and in boys when WHR was greater than 0.9 [51]. Blood pressure (BP) was measured using the auscultatory method with the use of a sphygmomanometer. Before these measurements, the subjects rested quietly for 5 min. The measurements were conducted in triplicate at 1–2 min intervals. Hypertension (HT) was defined when the mean value of systolic blood pressure (SBP) and/or diastolic blood pressure (DBP) was ≥ the 95th percentile, corrected for appropriate sex, age, and height [26].

### 4.2. Blood Collection

Blood samples were taken after an 8-hour overnight fast. The blood was collected into tubes containing a serum clot activator. The samples were centrifuged for 5 min at 4000 rpm, and then the obtained serum was transferred to new, clean tubes. The collected material was stored at −80 °C until further analysis.

### 4.3. Determination of Triacylglycerol Concentrations

The serum concentration of TAG was measured by the enzymatic–colorimetric method using commercial kits (Roche Diagnostics GmbH, Mannheim, Germany) with the Cobas 6000 c502 Roche (Hitachi High-Technologies Corporation, Tokyo, Japan), following the manufacturer’s protocol.

### 4.4. Determination of the Apolipoprotein Concentrations

The concentration of apolipoproteins—i.e., Apo-A1, Apo-A2, Apo-B, Apo-C1, Apo-C3, Apo-D, Apo-E, Apo-H, Apo-J—and CRP were measured using the Immunoassay Kit (Bio-Plex Pro Human Apolipoprotein 10-Plex Assay, Bio-Rad; Warsaw, Poland).

Before performing the multiplex assay, the sample dilutions were prepared following the manufacturer’s protocol. The serum samples were centrifuged at 1000× *g* for 15 min at 4 °C to remove particulates. Thereafter, the samples were diluted in three steps with sample dilution buffer (1:50,000) (Bio-Rad, Hercules, CA, USA). This multiplexing method is based on sensitive covalently coupled magnetic beads. Firstly, standards, blank, and samples were added to the appropriate wells of a 96-well assay microplate. Subsequently, the capture beads were applied to each well and incubated while protected from light for 1 h. Next, the microplate was washed three times using a diluted assay buffer. The detection antibodies were applied to each well, and then the plate was incubated again for 1 h. After addition of streptavidin–phycoerythrin (SA–PE) solution, subsequent incubation, and a series of successive washes, the resuspended beads were added to each well and shaken for 30 s. Finally, the levels of the selected parameters were determined using the Bio-Plex 200 System (Bio-Rad Laboratories, Inc.; Hercules, CA, USA). The Apo concentrations were calculated with consideration of the appropriate standard curve.

### 4.5. Statistical Analysis

The data are presented as mean ± standard deviations (SD). Statistical analysis was conducted using GraphPad Prism 8.2.1. software (San Diego, CA, USA). The normal distribution of the values was assessed by the Shapiro–Wilk test. Statistical comparisons were performed with the non-parametric Mann–Whitney U test (for non-normally distributed variables) or the parametric *t*-test test (for normally distributed variables). Spearman’s rank correlation coefficient was used to correlate results between selected parameters. Optimal cutoff points and determination of the diagnostic value of apolipoproteins were evaluated using the ROC method. The association between Apo levels and the independent variables was established using multivariate regression models. The results were considered statistically significant with a *p*-value < 0.05.

## 5. Conclusions

In conclusion, this study showed that patients with childhood ALL and history of anticancer treatment are at high risk of the development of alterations in their Apo profiles. Moreover, the composition of Apo-A1, Apo-A2, and Apo-D was discovered to be notably changed in ALL survivors compared to the normal-weight, healthy individuals. Furthermore, we found a significant correlation between increased BMI and elevated Apo-C1 concentration in patients with a history of anticancer treatment targeting ALL. It should be noted that this is the first study to reveal that apolipoproteins—specifically, Apo-A1, Apo-A2, Apo-C1, and Apo-D—might constitute valuable biomarkers of metabolic imbalance, which could be used for the long-term prognosis of the development of overweight and obesity among ALL survivors. The possibility that abnormal Apo composition may lead to the onset of MetS—including significant weight gain, hypertriglyceridemia, and hypercholesterolemia—cannot be excluded; therefore, systematic follow-up for this condition is necessary in this population. Considering the role of the changed Apo composition in the development of MetS among children after ALL treatment, we believe that this paper, at least partially, highlights the importance of long-term prognosis of the occurrence of metabolic complications that are directly associated with anticancer treatment for hematological malignancies.

## Figures and Tables

**Figure 1 ijms-23-10634-f001:**
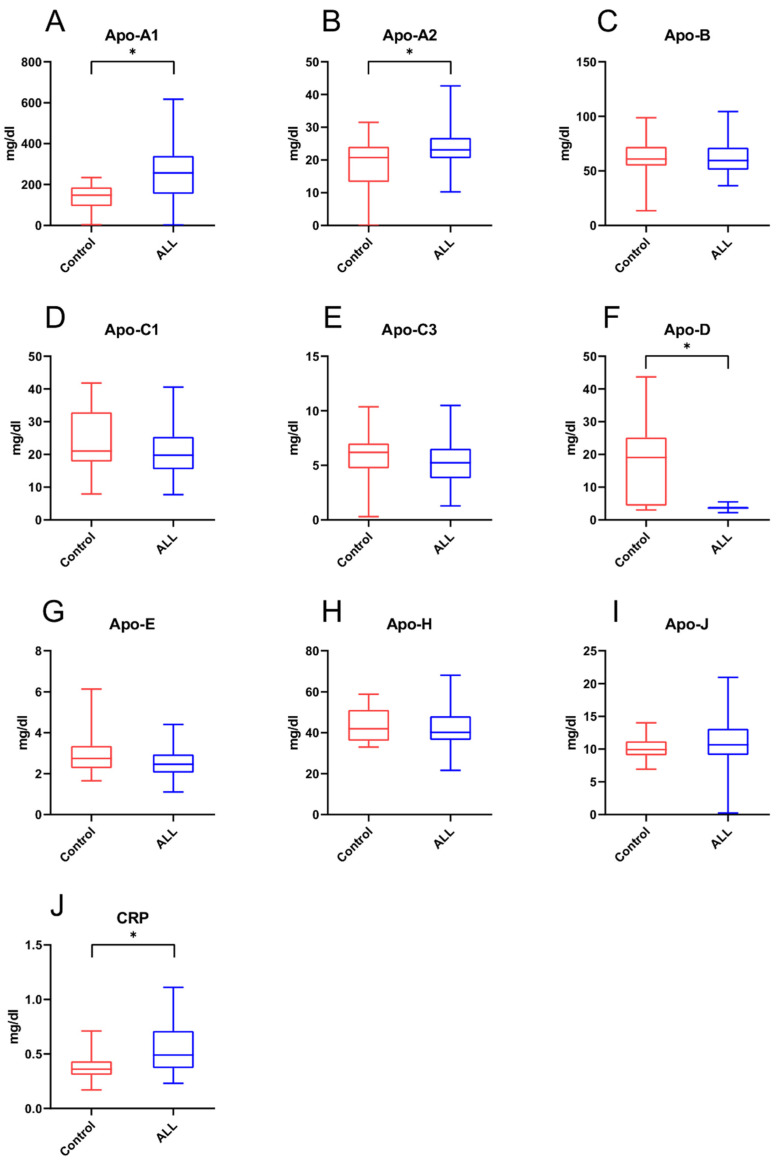
The comparison of apolipoprotein concentrations—i.e., (**A**) apolipoprotein A1, (**B**) apolipoprotein A2, (**C**) apolipoprotein B, (**D**) apolipoprotein C1, (**E**) apolipoprotein C3, (**F**) apolipoprotein D, (**G**) apolipoprotein E, (**H**) apolipoprotein H, (**I**) apolipoprotein J—and (**J**) C-reactive protein between acute lymphoblastic leukemia (ALL) survivors with normal and abnormal BMI and the control group. Data are presented as the mean ± standard deviation (SD). * *p* < 0.05.

**Figure 2 ijms-23-10634-f002:**
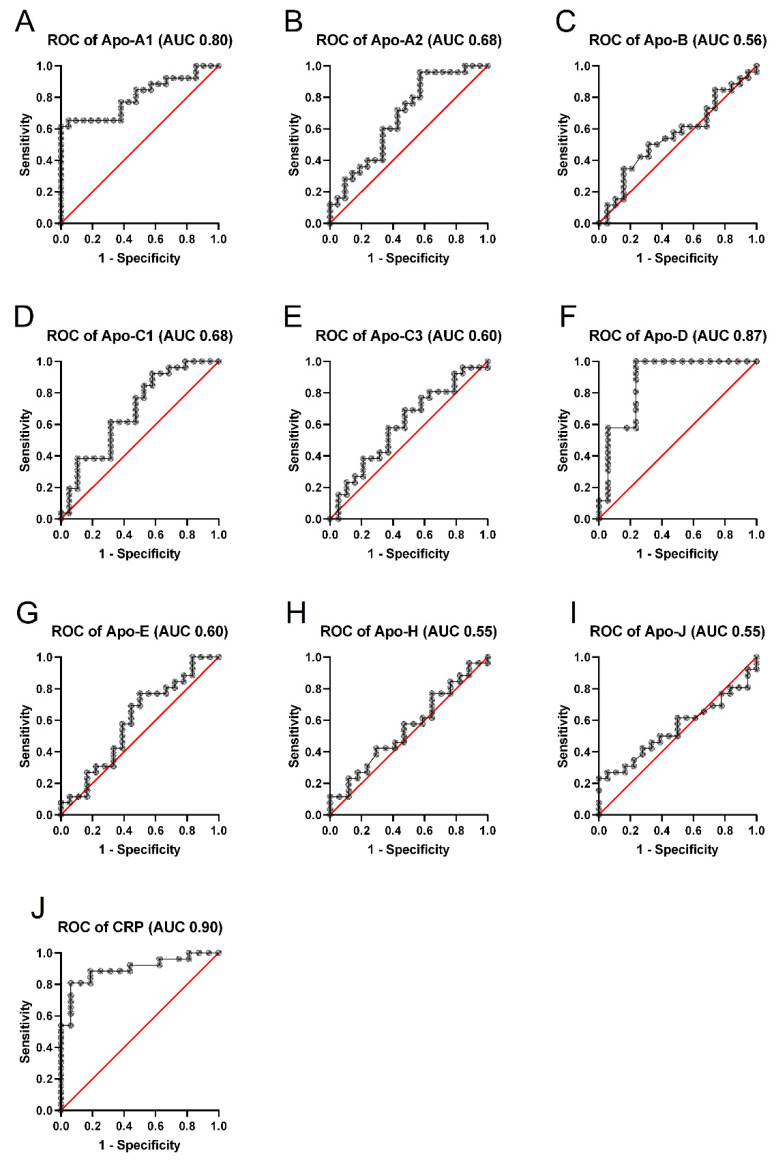
The receiver operating characteristic (ROC) analysis for the prediction of abnormal weight (i.e., overweight/obese) based on the concentrations of apolipoproteins—i.e., (**A**) apolipoprotein A1, (**B**) apolipoprotein A2, (**C**) apolipoprotein B, (**D**) apolipoprotein C1, (**E**) apolipoprotein C3, (**F**) apolipoprotein D, (**G**) apolipoprotein E, (**H**) apolipoprotein H, (**I**) apolipoprotein J—and (**J**) C-reactive protein in acute lymphoblastic leukemia (ALL) survivors.

**Table 1 ijms-23-10634-t001:** Baseline characteristics of the study group.

	Total Number (%) or Mean ± SD
Patients	58 (100%)
Sex	
Female	27 (46.55%)
Male	31 (53.45%)
Age at diagnosis (years)	5.01 ± 3.46
Age with treatment completion (years)	7.18 ± 3.31
Follow-up after treatment (years)	5.41 ± 4.29
Body Mass Index (BMI, kg/m^2^)	21.95 ± 5.29
Weight status	
Normal weight	31 (53.45%)
Overweight	17 (29.31%)
Obese	10 (17.24%)
Waist–Hip Ratio (WHR)	
Female	0.84 ± 0.06
Male	0.89 ± 0.07
Systolic Blood Pressure (SBP, mmHg)	109.10 ± 15.81
Diastolic Blood Pressure (DBP, mmHg)	67.30 ± 12.02
Anticancer treatment	
Hematopoietic Stem-Cell Transplantation (HSCT)	6 (10.34%)
Radiotherapy (RT)	9 (15.52%)
Cranial Radiotherapy (CRT)	8 (13.79%)
Total Body Irradiation (TBI)	2 (3.45%)
CRT and TBI	1 (1.72%)
Non-Radiotherapy (Non-RT)	49 (84.48%)
Cytostatic agents	58 (100%)
Cyclophosphamide (cumulative dose in mg/m^2^)	58 (100%); 3957.00 ± 2633.00
Methotrexate (cumulative dose in mg/m^2^)	58 (100%); 10,321.00 ± 6644.00
Glucocorticoids	58 (100%)
Cumulative corticosteroid dose (mg/m^2^) ^a^	58 (100%); 3538.00 ± 901.80
Dexamethasone (cumulative dose in mg/m^2^)	58 (100%); 277.30 ± 134.60
Prednisone (cumulative dose in mg/m^2^)	58 (100%); 1680.00 ± 0.00

^a^ Calculated as prednisone equivalents.

**Table 2 ijms-23-10634-t002:** The comparison of apolipoprotein concentrations in ALL survivors with normal weight, ALL survivors with abnormal weight, and control patients with normal weight.

	Groups and Subgroups	Values	95% Confidence Interval of Area under the ROC Curve, and *p*-Value		
			Normal Weight (ALL Group) vs. Overweight and Obese (ALL Group)	Normal Weight (ALL Group) vs. Normal Weight (Control Group)	Overweight and Obese (ALL Group) vs. Normal Weight (Control Group)
Apo-A1 (mg/dL)	Normal weight—ALL group	263.93 ± 139.68	0.36–0.66 0.911	0.65–0.90 0.001	0.68–0.93 0.001
	Overweight and obese—ALL group	254.32 ± 108.86			
	Normal weight—control group	135.25 ± 67.16			
Apo-A2 (mg/dL)	Normal weight—ALL group	25.34 ± 6.07	0.39–0.70 0.560	0.58–0.86 0.007	0.52–0.84 0.040
	Overweight and obese—ALL group	24.04 ± 6.80			
	Normal weight—control group	18.00 ± 8.78			
Apo-B (mg/dL)	Normal weight—ALL group	61.55 ± 14.75	0.36–0.67 0.863	0.37–0.70 0.666	0.39–0.73 0.512
	Overweight and obese—ALL group	60.69 ± 15.28			
	Normal weight—control group	61.26 ± 17.19			
Apo-C1 (mg/dL)	Normal weight—ALL group	22.35 ± 6.97	0.52–0.81 0.034	0.39–0.74 0.442	0.52–0.85 0.036
	Overweight and obese—ALL group	18.37 ± 5.59			
	Normal weight—control group	24.31 ± 9.04			
Apo-C3 (mg/dL)	Normal weight—ALL group	5.36 ± 1.81	0.35–0.66 0.934	0.44–0.78 0.196	0.43–0.77 0.260
	Overweight and obese—ALL group	5.30 ± 2.13			
	Normal weight—control group	6.00 ± 2.37			
Apo-D (mg/dL)	Normal weight—ALL group	3.63 ± 0.72	0.38–0.68 0.683	0.78–1.00 0.001	0.78–1.00 0.001
	Overweight and obese—ALL group	3.60 ± 0.61			
	Normal weight—control group	28.74 ± 28.90			
Apo-E (mg/dL)	Normal weight—ALL group	2.82 ± 1.31	0.35–0.66 0.948	0.41–0.74 0.394	0.42–0.77 0.283
	Overweight and obese—ALL group	2.58 ± 0.76			
	Normal weight—control group	3.03 ± 1.21			
Apo-H (mg/dL)	Normal weight—ALL group	41.41 ± 11.62	0.38–0.69 0.608	0.34–0.70 0.803	0.34–0.70 0.830
	Overweight and obese—ALL group	41.73 ± 9.06			
	Normal weight—control group	41.04 ± 12.68			
Apo-J (mg/dL)	Normal weight—ALL group	11.45 ± 3.70	0.42–0.73 0.353	0.52–0.82 0.045	0.40–0.74 0.415
	Overweight and obese—ALL group	10.69 ± 3.27			
	Normal weight—control group	9.59 ± 2.88			
CRP (mg/dL)	Normal weight—ALL group	0.58 ± 0.38	0.62–0.88 0.001	0.45–0.79 0.180	0.65–0.95 0.001
	Overweight and obese—ALL group	1.17 ± 1.10			
	Normal weight—control group	0.65 ± 0.81			

**Table 3 ijms-23-10634-t003:** Characteristics of patients with acute lymphoblastic leukemia (ALL) according to age at diagnosis. Data are presented as the mean ± standard deviation (SD).

	<6 Years—Age at Diagnosis of ALL *n* = 39	>6 Years—Age at Diagnosis of ALL *n* = 19	*p*-Value
TAG (mg/dL)	95.93 ± 41.10	101.60 ± 63.39	0.860
Apo-A1 (mg/dL)	267.27 ± 115.99	219.50 ± 119.85	0.175
Apo-A2 (mg/dL)	24.87 ± 6.12	22.96 ± 6.31	0.439
Apo-B (mg/dL)	61.06 ± 14.91	59.77 ± 15.50	0.778
Apo-C1 (mg/dL)	20.20 ± 6.18	19.80 ± 7.02	0.836
Apo-C3 (mg/dL)	5.10 ± 1.65	5.46 ± 2.30	0.528
Apo-D (mg/dL)	3.59 ± 0.49	3.51 ± 0.88	0.851
Apo-E (mg/dL)	2.72 ± 1.14	2.57 ± 0.98	0.848
Apo-H (mg/dL)	41.17 ± 8.24	40.35 ± 13.49	0.670
Apo-J (mg/dL)	10.91 ± 2.62	10.97 ± 4.67	0.952
CRP (mg/dL)	0.84 ± 0.94	0.91 ± 0.61	0.142

**Table 4 ijms-23-10634-t004:** Spearman’s rank correlation of apolipoprotein concentrations in patients with acute lymphoblastic leukemia (ALL), by weight status.

	Variable	Spearman’s r	*p*-Value	95% Confidence Interval
Apo-A1	Normal weight	0.01	0.965	−0.44–0.46
	Overweight and obese	0.22	0.282	−0.19–0.57
Apo-A2	Normal weight	−0.29	0.221	−0.67–0.20
	Overweight and obese	0.30	0.145	−0.12–0.63
Apo-B	Normal weight	−0.01	0.977	−0.47–0.46
	Overweight and obese	0.31	0.117	−0.09–0.63
Apo-C1	Normal weight	−0.59	0.008	−0.83–−0.17
	Overweight and obese	0.26	0.203	−0.15–0.59
Apo-C3	Normal weight	−0.07	0.781	−0.52–0.41
	Overweight and obese	0.60	0.001	0.27–0.81
Apo-D	Normal weight	−0.04	0.867	−0.48–0.42
	Overweight and obese	0.56	0.003	0.21–0.78
Apo-E	Normal weight	−0.35	0.145	−0.70–0.14
	Overweight and obese	0.37	0.063	−0.03–0.67
Apo-H	Normal weight	−0.34	0.144	−0.69–0.14
	Overweight and obese	0.49	0.010	0.12–0.74
Apo-J	Normal weight	−0.07	0.757	−0.51–0.39
	Overweight and obese	0.41	0.037	0.01–0.69
CRP	Normal weight	0.42	0.072	−0.05–0.74
	Overweight and obese	0.47	0.016	0.09–0.73

**Table 5 ijms-23-10634-t005:** Spearman’s rank correlation of apolipoprotein concentrations in patients with acute lymphoblastic leukemia (ALL), depending on the selected anticancer chemotherapy.

Variable		Spearman’s r	*p*-Value	95% Confidence Interval
Cyclophosphamide (cumulative dose in mg/m^2^)	Apo-A1	0.06	0.648	−0.21–0.33
	Apo-A2	−0.01	0.979	−0.28–0.27
	Apo-B	0.07	0.599	−0.20–0.34
	Apo-C1	0.07	0.602	−0.21–0.34
	Apo-C3	0.07	0.613	−0.21–0.34
	Apo-D	0.08	0.557	−0.20–0.35
	Apo-E	0.08	0.576	−0.20–0.34
	Apo-H	0.08	0.543	−0.19–0.35
	Apo-J	0.02	0.853	−0.25–0.30
	CRP	0.13	0.356	−0.15–0.39
Methotrexate (cumulative dose in mg/m^2^)	Apo-A1	0.09	0.499	−0.18–0.36
	Apo-A2	−0.01	0.952	−0.28–0.27
	Apo-B	−0.05	0.701	−0.32–0.22
	Apo-C1	0.10	0.459	−0.18–0.37
	Apo-C3	0.05	0.715	−0.23–0.32
	Apo-D	0.08	0.574	−0.20–0.34
	Apo-E	0.22	0.103	−0.05–0.47
	Apo-H	0.04	0.747	−0.23–0.31
	Apo-J	0.13	0.359	−0.15–0.38
	CRP	0.26	0.060	−0.02–0.50
Cumulative corticosteroid dose (mg/m^2^)	Apo-A1	0.13	0.357	−0.15–0.39
	Apo-A2	−0.07	0.637	−0.34–0.21
	Apo-B	−0.04	0.764	−0.31–0.24
	Apo-C1	0.01	0.971	−0.27–0.28
	Apo-C3	0.01	0.974	−0.27–0.28
	Apo-D	0.12	0.387	−0.16–0.38
	Apo-E	−0.01	0.989	−0.28–0.27
	Apo-H	0.09	0.517	−0.19–0.35
	Apo-J	0.10	0.458	−0.17–0.36
	CRP	0.12	0.389	−0.16–0.38
Dexamethasone (cumulative dose in mg/m^2^)	Apo-A1	0.13	0.357	−0.15–0.39
	Apo-A2	−0.07	0.637	−0.34–0.21
	Apo-B	−0.04	0.764	−0.31–0.24
	Apo-C1	0.01	0.971	−0.27–0.28
	Apo-C3	0.01	0.974	−0.27–0.28
	Apo-D	0.12	0.387	−0.16–0.38
	Apo-E	−0.01	0.988	−0.28–0.27
	Apo-H	0.09	0.517	−0.19–0.35
	Apo-J	0.10	0.458	−0.17–0.36
	CRP	0.12	0.389	−0.16–0.38

## Data Availability

Not applicable.

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
