# Peer review of "Apolipoproteins—New Biomarkers of Overweight and Obesity among Childhood Acute Lymphoblastic Leukemia Survivors?"

_ijms, 2022, doi:10.3390/ijms231810634_

Round 1

Reviewer 1 Report

In this article, the authors have studied and identified biomarkers for the occurrence of obesity in children who have undergone treatment for ALL through an extensive panel profiling of apolipoproteins. Through a multitude of analyses, the authors determined that;

a.       TAG levels are comparatively at elevated levels in ALL survivors with an unhealthy weight as compared to those bearing a healthy weight upon analysis by BMI

b. By comparing the concentrations of different Apo between the ALL study groups versus the control group, the authors identified increased concentrations of Apo-A1, Apo-A2, and CRP but a significant decrease in ApoD in the ALL study participants with respect to the control participants.

c.       AOC analysis to predict obesity occurrence in the ALL survivors, Apo-D had the biggest vale and hence was suggested as a biomarker for obesity.

In conclusion, the authors identify a panel of apolipoproteins resulting from the anticancer therapy in ALL survivors which may serve as a biomarker for a prediction for overweight and obesity as an after-treatment adverse effect in these individuals.

Page 6: AOC analysis- Do the authors not consider CRP as another marker based on the AOC? Further CRP seems to consistently bear a difference with respect to the control group with the analysis of the concentration as well as with BMI relation when compared within the ALL survivor group.

Another major limitation of the study is incorporating n overweight/obesity-afflicted control group for proper comparison of the changes seen in the apolipoproteins with specific respect to the therapy received by the individuals in the ALL group as mentioned by the authors in their discussion.

Minor suggestions:

Page 4, Line 109:  maybe reworded as: “The analysis by BMI…. TAG level was increased in ALL survivors with unhealthy weight as compared to those with a healthy weight …..”

Page 4, Line 133: may be reworded to “Additionally, CRP concentration was considerably increased in childhood cancer survivors with respect to the control group…”

Page 6, Line 162: Please replace “Mentioned results” with “Presented results”

Page 10, Line 244: Suggestion to replace “whole AAL” with “ALL group in the entirety”

Page 10, Line 248: Suggestion to replace “children with ALL in the history” with “children with a history of ALL affliction”

Page 10, Line 267: “Steam” should be “stem”

Page 10, Line 268:  May be reworded to “who underwent cranial radiation with an insufficient period of follow-up period”

Page 10, Line 270: “Could” should be replaced by “Can”  

Page 10, Line 277: Please replace “likewise” with “as well as” or “and”

Page 10, Line 285: “Some authors” should be worded as “it has been previously reported that”

Recommendation for authors to have Abbreviations for the Apo be mentioned at the start of the paper (best placed after the abstract) rather than having to mention it after every figure.

Author Response

Białystok, 2022-08-30

Dear Madam, Dear Sir,

We appreciate the time and effort that you dedicated to providing your valuable feedback on our manuscript entitled “Apolipoproteins – new biomarkers of overweight and obesity among childhood acute lymphoblastic leukemia survivors?” (authors: Klaudia Sztolsztener, Hubert Å»ywno, Katarzyna Hodun, Katarzyna KonoÅ„czuk, Katarzyna MuszyÅ„ska-RosÅ‚an, Eryk Latoch). We are grateful for your insightful comments on our paper. Furthermore, the whole manuscript was corrected and improved. The changes within the manuscript have been highlighted.

Comments and Suggestions for Authors

In this article, the authors have studied and identified biomarkers for the occurrence of obesity in children who have undergone treatment for ALL through an extensive panel profiling of apolipoproteins. Through a multitude of analyses, the authors determined that;

  1. TAG levels are comparatively at elevated levels in ALL survivors with an unhealthy weight as compared to those bearing a healthy weight upon analysis by BMI
  2. By comparing the concentrations of different Apo between the ALL study groups versus the control group, the authors identified increased concentrations of Apo-A1, Apo-A2, and CRP but a significant decrease in ApoD in the ALL study participants with respect to the control participants.
  3. AOC analysis to predict obesity occurrence in the ALL survivors, Apo-D had the biggest vale and hence was suggested as a biomarker for obesity.

In conclusion, the authors identify a panel of apolipoproteins resulting from the anticancer therapy in ALL survivors which may serve as a biomarker for a prediction for overweight and obesity as an after-treatment adverse effect in these individuals.

Page 6: AOC analysis- Do the authors not consider CRP as another marker based on the AOC? Further CRP seems to consistently bear a difference with respect to the control group with the analysis of the concentration as well as with BMI relation when compared within the ALL survivor group.

Authors: Thank you for your valuable suggestion. We decided to not consider CRP as a prognostic marker because CRP is general and unspecific marker of inflammation. Inflammation state is a result of obesity however, it is not directly linked with metabolic imbalance as apolipoproteins seem to be. ALL itself is related to the development of inflammatory state and activation of leukocytes. Therefore, we decided to not include CRP in our analysis to not distract the reader attention and make our study too vast and incohesive.

Another major limitation of the study is incorporating n overweight/obesity-afflicted control group for proper comparison of the changes seen in the apolipoproteins with specific respect to the therapy received by the individuals in the ALL group as mentioned by the authors in their discussion.

Minor suggestions:

Page 4, Line 109:  maybe reworded as: “The analysis by BMI…. TAG level was increased in ALL survivors with unhealthy weight as compared to those with a healthy weight …..”

Page 4, Line 133: may be reworded to “Additionally, CRP concentration was considerably increased in childhood cancer survivors with respect to the control group…”

Page 6, Line 162: Please replace “Mentioned results” with “Presented results”

Page 10, Line 244: Suggestion to replace “whole AAL” with “ALL group in the entirety”

Page 10, Line 248: Suggestion to replace “children with ALL in the history” with “children with a history of ALL affliction”

Page 10, Line 267: “Steam” should be “stem”

Page 10, Line 268:  May be reworded to “who underwent cranial radiation with an insufficient period of follow-up period”

Page 10, Line 270: “Could” should be replaced by “Can”  

Page 10, Line 277: Please replace “likewise” with “as well as” or “and”

Page 10, Line 285: “Some authors” should be worded as “it has been previously reported that”

Authors: Thank you for all your suggestions. We corrected as you suggested.

Recommendation for authors to have Abbreviations for the Apo be mentioned at the start of the paper (best placed after the abstract) rather than having to mention it after every figure.

Authors: Abbreviations added.

We believe that the manuscript was further improved and is now more suitable for publication.

Yours faithfully,

Klaudia Sztolsztener

Department of Physiology,

Medical University of Białystok,

15-222 Białystok, Mickiewicz Str. 2C, Poland

Email: klaudia.sztolsztener@umb.edu.pl

Telephone: +48857485587

FAX: + 48857485586

Reviewer 2 Report

1. It would be better if you write for example apolipoprotein C1 (Apo-C1) once and then continue with abbreviations Apo-C1 in the rest of the manuscript.

2. You write that the healthy volunteers had normal body weight and proper body mass index and that they did not receive any medication at the time of the study. Wouldn´t it be more accurate to have one control group that is overweight but does not receive any medication, to compare with the overweight patients who recovered from ALL?

3. Is it possible that different therapies e.g. transplant, and radiation of different parts of the body could affect the concentration of Apolipoprotein and CRP etc?

4. Why didn´t you separate the groups by gender? Or maybe you couldn´t see any difference between males and females?

5. Just a comment: Apolipoprotein serum levels have previously been shown to be related to metabolic syndrome in patients with other diseases. 

Author Response

Białystok, 2022-08-30

Dear Madam, Dear Sir,

We appreciate the time and effort that you dedicated to providing your valuable feedback on our manuscript entitled “Apolipoproteins – new biomarkers of overweight and obesity among childhood acute lymphoblastic leukemia survivors?” (authors: Klaudia Sztolsztener, Hubert Å»ywno, Katarzyna Hodun, Katarzyna KonoÅ„czuk, Katarzyna MuszyÅ„ska-RosÅ‚an, Eryk Latoch). We are grateful for your insightful comments on our paper. Furthermore, the whole manuscript was corrected and improved. The changes within the manuscript have been highlighted.

Comments and Suggestions for Authors

  1. It would be better if you write for example apolipoprotein C1 (Apo-C1) once and then continue with abbreviations Apo-C1 in the rest of the manuscript.

Authors: Thank you for this comment. In the main text we replaced full names for apolipoprotein and others by using their abbreviations. We also added abbreviations list into the manuscript. In some places of the manuscript, we have left the full names, due to journal requirements.

  1. You write that the healthy volunteers had normal body weight and proper body mass index and that they did not receive any medication at the time of the study. Wouldn´t it be more accurate to have one control group that is overweight but does not receive any medication, to compare with the overweight patients who recovered from ALL?

Authors: You are right that control group – volunteers with normal and also abnormal (overweight and obese) weight would enrich our research, which is an idea for the future research. This is a valuable suggestion. We will incorporate this type of group during the planning of the next study. In the herein study, we only focused on changes in Apo profile in ALL survivors in relation to the volunteers with normal weight. However, in the last paragraph in the Discussion section we described several limitations of present study, e.g., no patients with abnormal body weight in the control group.

  1. Is it possible that different therapies e.g. transplant, and radiation of different parts of the body could affect the concentration of Apolipoprotein and CRP etc?

Authors: Yes, it is possible, so in the herein study we examined the correlation of Apo concentration depending on different treatment factors in the history of ALL survivors. Our results are presented in Table 5 and the description in point 2.7. in the Results section. Available data showed that radiotherapy and hematopoietic stem cell transplantation affect Apo and CRP levels in cancer patients. Thus, we determined to what extent Apo and CRP concentrations change in the serum of ALL survivors after treatment with and without radiotherapy.

  1. Why didn´t you separate the groups by gender? Or maybe you couldn´t see any difference between males and females?

Authors: In the Table S1 in Supplementary Materials we presented the relationship in the concentration of Apo in females and males among ALL survivors. Statistical analysis did not show any differences in the content of examined parameters depending on gender.

  1. Just a comment: Apolipoprotein serum levels have previously been shown to be related to metabolic syndrome in patients with other diseases. 

We believe that the manuscript was further improved and is now more suitable for publication.

Yours faithfully,

Klaudia Sztolsztener

Department of Physiology,

Medical University of Białystok,

15-222 Białystok, Mickiewicz Str. 2C, Poland

Email: klaudia.sztolsztener@umb.edu.pl

Telephone: +48857485587

FAX: + 48857485586

Reviewer 3 Report

In the present study authors address the important problem of metabolic complications in survivors of acute lymphoblastic leukemia. In particular, the levels of apolipoproteins in childhood ALL survivors were measured in 58 patients in complete remission as well as in 22 control subjects. All patients were treated with cytotoxic drugs and glucocorticoids and 9 of them received additional radiotherapy. The main findings are as follows: (1) age of ALL diagnosis has no effect on apolipoprotein concentrations in ALL survivors, (2) apo-C1 is lower and CRP is higher in overweight/obese than in normal weight survivors, (3) triglycerides, apo-C3 apo-J tended to be lower in patients without radiotherapy than in those treated with radiotherapy, (4) apo A1, apo-A2 and CRP were higher in ALL survivors than in control subjects whereas apo-D was markedly lower in ALL group.

The topic and the results are of interest, however, the results are mostly descriptive and clinical implications of the findings are not clear.

Specific comments:

1)     Line 67: “apo-2” should be corrected to: “apo-A2”.

2)     Line 312: “receiving by patients” should be corrected to: “received by patients”.

3)     Line 348: the name of the manufacturer of triglyceride measurement kit should be specified.

4)     Authors could consider to change the order of data presentation. In particular, according to the aim of the study, apolipoprotein concentrations should be first compared between control and ALL groups.

5)     Results concerning the effect of body weight (Table 3 and section 2.4) should be presented together in one section.

6)     Line 146: do the authors mean correlation with normal weight or correlation with body weight?

7)     Table 4: why correlation with body weight was calculated separately for normal weight and overweight/obese subgroups?

8)     Why apolipoproteins are considered as the predictors of obesity? Obesity is easily diagnosed according to body weight measurement and differences in apolipoprotein levels are the consequence rather than the cause of obesity.

9)     Why other components of the lipid profile (e.g. LDL- and HDL-cholesterol) were not measured together with triglycerides?

Author Response

Białystok, 2022-08-30

Dear Madam, Dear Sir,

We appreciate the time and effort that you dedicated to providing your valuable feedback on our manuscript entitled “Apolipoproteins – new biomarkers of overweight and obesity among childhood acute lymphoblastic leukemia survivors?” (authors: Klaudia Sztolsztener, Hubert Å»ywno, Katarzyna Hodun, Katarzyna KonoÅ„czuk, Katarzyna MuszyÅ„ska-RosÅ‚an, Eryk Latoch). We are grateful for your insightful comments on our paper. Furthermore, the whole manuscript was corrected and improved. The changes within the manuscript have been highlighted.

Comments and Suggestions for Authors

In the present study authors address the important problem of metabolic complications in survivors of acute lymphoblastic leukemia. In particular, the levels of apolipoproteins in childhood ALL survivors were measured in 58 patients in complete remission as well as in 22 control subjects. All patients were treated with cytotoxic drugs and glucocorticoids and 9 of them received additional radiotherapy. The main findings are as follows: (1) age of ALL diagnosis has no effect on apolipoprotein concentrations in ALL survivors, (2) apo-C1 is lower and CRP is higher in overweight/obese than in normal weight survivors, (3) triglycerides, apo-C3 apo-J tended to be lower in patients without radiotherapy than in those treated with radiotherapy, (4) apo A1, apo-A2 and CRP were higher in ALL survivors than in control subjects whereas apo-D was markedly lower in ALL group.

The topic and the results are of interest, however, the results are mostly descriptive and clinical implications of the findings are not clear.

Specific comments:

1) Line 67: “apo-2” should be corrected to: “apo-A2”.

Authors: According to your suggestion, we improved.

2) Line 312: “receiving by patients” should be corrected to: “received by patients”.

Authors: According to your suggestion, we improved.

3) Line 348: the name of the manufacturer of triglyceride measurement kit should be specified.

Authors: We added more information.

4) Authors could consider to change the order of data presentation. In particular, according to the aim of the study, apolipoprotein concentrations should be first compared between control and ALL groups.

Authors: Thank you for suggestion. We changed order the results presentation. In the Results section, the basic characteristic of patients is the first, then the comparison of Apo panel between ALL and Control groups, and finally, precise characteristic of Apo concentration in ALL group.

5) Results concerning the effect of body weight (Table 3 and section 2.4) should be presented together in one section.

Authors: We changed it according to your suggestion.

6) Line 146: do the authors mean correlation with normal weight or correlation with body weight?

Authors: This correlation exists between Apo-C1 levels and only normal weight in ALL survivors.

7) Table 4: why correlation with body weight was calculated separately for normal weight and overweight/obese subgroups?

Authors: In our study, the results from Table 4 are continuation of results presented in Table 3, which shows the exact differences in Apo concentrations in the normal weight and overweight/obese subgroups among ALL survivors. In Table 3, we checked whether the Apo concentrations in the groups with normal weight and overweight/obese show statistically significant differences. In order to continue the statistical analysis, Table 4 presents the results of the analysis aimed at determining the exact correlation between the studied parameters (negative or positive correlation) within the subgroups (normal and abnormal body weight) with the exact BMI values of the patients. In our research we show that mentioned relationships exist and perhaps the changes in lipid metabolism may lead to abnormal body weight. Therefore we decided to show this correlation depending on the weight status – separately for both subgroups - normal and abnormal weight as a long-term predictors of abnormal weight.

8) Why apolipoproteins are considered as the predictors of obesity? Obesity is easily diagnosed according to body weight measurement and differences in apolipoprotein levels are the consequence rather than the cause of obesity.

Authors: We agree with the reviewer's opinion that obesity is easily diagnosed on the basis of anthropometric measurements. However, it has not yet been elucidated whether anticancer treatment can alter lipid metabolism and then promoting the development of long-term metabolic complications. It is difficult to draw unequivocal conclusions about the predictive factors on the basis of the 'cross-sectional study'. However, in our study we show that mentioned relationships exist and perhaps the changes in lipid metabolism may lead to abnormal body weight. But, this requires further investigation. We believe that our results will contribute to a better understanding of this problem in the future.

9) Why other components of the lipid profile (e.g. LDL- and HDL-cholesterol) were not measured together with triglycerides?

Authors: We have only shown the TAG level in our manuscript. Unfortunately, due to the very small amount of material obtained from ALL patients, the concentrations of other parameters from lipid profile were determined only in a few patients. Due to the fact that we have a small number of these results, we decided not to present them in the herein study. Additionally, in the Discussion section, we added information about this limitation.

We believe that the manuscript was further improved and is now more suitable for publication.

Yours faithfully,

Klaudia Sztolsztener

Department of Physiology,

Medical University of Białystok,

15-222 Białystok, Mickiewicz Str. 2C, Poland

Email: klaudia.sztolsztener@umb.edu.pl

Telephone: +48857485587

FAX: + 48857485586
